The effects of gluteal squeezes compared to bilateral bridges on gluteal strength, power, endurance, and girth

Lehecka Bryan J. bryan.lehecka@wichita.edu
Turley Jessica
Stapleton Aaron
Waits Kyle
Zirkle John
Department of Physical Therapy, Wichita State University , Wichita, KS , USA
Abdala Virginia
Electronic publication date: 2019 Jul 8
Publication date: 2019
Volume: 7
Electronic Location ID: e7287
Received 2019 Mar 4; Accepted 2019 Jun 12
Copyright: © 2019 Lehecka et al.
Copyright year: 2019
Copyright holder: Lehecka et al.
License: This is an open access article distributed under the terms of the Creative Commons Attribution License, which permits unrestricted use, distribution, reproduction and adaptation in any medium and for any purpose provided that it is properly attributed. For attribution, the original author(s), title, publication source (PeerJ) and either DOI or URL of the article must be cited.
License URL: https://creativecommons.org/licenses/by/4.0/

Keywords: Gluteals, Hip, Strength, Power, Endurance, Girth

Funding: The authors received no funding for this work.

==============================
Background

Hip extension weakness is correlated with low back, hip, and knee pathology. Isometric gluteal squeezes have been shown to elicit high electromyographic gluteal activity. However, there is little research regarding the specific effects of isometric gluteal squeezes on hip strength and functional outcomes. The purpose of this study was to identify the effects of gluteal squeezes on hip extension strength, vertical jump, broad jump, single-leg bridge endurance, and gluteal girth compared to bilateral gluteal bridging.

Methods

A total of 32 healthy university students (mean age 23.28 ± 2.15 years) were randomly assigned to perform either gluteal squeezes or bilateral bridges daily. Subjects were tested at baseline and after 8 weeks of training. Subjects’ hip extension strength, vertical jump, broad jump, single-leg bridge endurance, and gluteal girth were tested.

Results

No statistically significant differences were found between the bridge and squeeze groups after 8 weeks of training. Both groups significantly improved hip extension strength bilaterally (p = 0.000–0.011). The squeeze group significantly increased gluteal girth at the level of the greater trochanter (p = 0.007), but no significant girth increase was seen in the bridge group (p = 0.742). Although increases were seen in both groups for the endurance and jump tasks, no statistically significant changes occurred for those outcomes. All outcome measurements demonstrated high reliability (ICC = 0.93–0.99).

Conclusion

Gluteal squeezes were as effective as bilateral bridges for increasing hip extension strength. Gluteal squeezes also significantly increased girth at the level of the greater trochanter. These results provide clinical and aesthetic reasons to perform gluteal squeezes.

Introduction

The gluteal squeeze is a convenient, isometric exercise shown to produce higher electromyographic (EMG) activity than many traditional therapeutic exercises (Boren et al., 2011). In fact, the standing gluteal squeeze elicits sufficient EMG activity to serve as a position by which to achieve a maximum voluntary isometric contraction of the gluteus maximus (Contreras et al., 2015). Despite the convenience and high gluteal activity produced by the gluteal squeeze exercise, its effects on gluteal strength, power, endurance, and girth have yet to be examined.

Gluteal weakness has been associated with multiple types of injuries, and gluteal strengthening has been linked to successful rehabilitation from multiple pathologies. For example, gluteal strength (hip abduction, external rotation, and extension strength) has independently been linked to non-contact anterior cruciate ligament injuries and associated with knee overuse injuries (Khayambashi et al., 2016; Kollock, Andrews & Johnston, 2016). Subjects with chronic low back pain and those with sacroiliac dysfunction reported decreased pain after gluteal strengthening (Added et al., 2018; Park & Lee, 2016). Gluteal strength has also been linked to falls in an elderly population (Inacio et al., 2014) and medial tibial pain in college-aged females (Verrelst et al., 2014). Gluteus maximus strength (hip extension and external rotation), specifically, has been linked to patellofemoral pain in multiple studies (Thomson et al., 2016; Van Cant et al., 2014) while the relationship between gluteus medius strength and patellofemoral pain seems more unclear (Neal et al., 2019). It appears, therefore, that gluteal strength is associated with injuries and pathology from the low back to the lower leg.

Gluteal power and endurance appear to be important constructs distinct from gluteal strength (Ambegaonkar et al., 2014). One study concluded hip extension endurance is significantly correlated to patellofemoral pain and altered kinematics during running (Souza & Powers, 2009). Another study demonstrated that hip extension power is linked to vertical jump height (Ford et al., 2009). Evidence suggests gluteal power and endurance are significant facets of function separate from gluteal strength.

Gluteal girth is a less researched feature of the muscles compared to their strength, power, and endurance, but noteworthy nonetheless. Muscle hypertrophy can be a result of strength or endurance training (Schoenfeld et al., 2015). Surgical augmentation is increasing in popularity as a means to increase gluteal girth, but it carries complication rates between 10% and 22% (Sinno et al., 2016). To the authors’ knowledge, no studies have examined the effect of a specific exercise on gluteal girth despite gluteal girth being a desired anthropometric characteristic (Singh, 1994).

Evidence about the effects of specific therapeutic exercises on gluteal strength, power, endurance, and girth is limited. Moreover, the effects of gluteal squeezes have not been studied despite the exercise producing especially high gluteal EMG activity. Similar to the standing gluteal squeeze, no evidence exists about the outcomes of seated gluteal squeezes, arguably the simplest and most convenient form of the exercise. While increasing degrees of hip flexion may lower gluteal activation compared to full hip extension (Worrell et al., 2001), a seated exercise is expected to lead to increased exercise compliance. There is a high rate of non-adherance to home exercise programs prescribed in physical therapy (Sluijs, Kok & Van Der Zee, 1993). The problem most frequently reported from these patients is either that they lacked the time to exercise or the exercise did not fit into their daily routine. The primary purpose of this study is to examine the effects of gluteal squeezes compared to a traditional therapeutic exercise designed to strengthen the gluteals (bilateral bridges) on gluteal strength, power, endurance, and girth. A secondary purpose of this study is to compare exercise compliance between subjects prescribed gluteal squeezes and those assigned to supine bilateral bridges. Gluteal squeezes are expected to produce significant increases in all outcomes to the same or higher degree than supine bilateral bridges. Subjects’ exercise compliance is expected to be higher among the squeeze group than the bridge group.

Methods

A total of 32 subjects (17 males, 15 females) were recruited for this mixed design study from a university population following Wichita State University Institutional Review Board approval (#3875). Healthy subjects between 18 and 35 years of age completed a written informed consent form and health questionnaire prior to testing and were excluded if they reported any of the following: orthopedic surgery within the last 6 months; current pregnancy; or current spine, hip, or knee pain. Subjects were assigned to either the gluteal squeeze group or bilateral bridge group using a random draw without replacement method. Upon group assignment, subjects were educated on their associated 8-week exercise program. Subjects also received an email specific to the group to which they were assigned demonstrating and explaining the proper exercise form and frequency.

Subjects completed a 2-min stationary bicycle warm-up between 30 and 60 rotations per minute. During the warm-up, subjects were educated on testing procedures and technique. After the warm-up, girth measurements with a standard measuring tape were taken at the level of the greater trochanter, five cm above the greater trochanter, and five cm below the greater trochanter while subjects stood relaxed with feet together. Subjects were asked to wear light, non-bulky athletic clothing for initial measurements, and identical clothing for post-intervention measurements. Hip girth measured similarly with a tape measure demonstrated high reliability (ICC = 0.96–0.97) in a previous study (Barrios et al., 2016). Next, gluteal strength was measured with a hand-held dynamometer (FEI Lafayette Manual Muscle Tester; Fig. 1) by positioning subjects in prone and asking them to perform hip extension against a dynamometer stabilized by a belt around the testing table with the knee bent to 90°. The peak force over 3 s was recorded. Subjects completed three trials per leg, alternating legs, and allowing at least 30 s of rest between trials. This method of strength testing has demonstrated high reliability (ICC = 0.91–0.94) in a previous study (Martins et al., 2017). Following these measurements, subjects’ vertical jump, broad jump, and timed single-leg bridge measurements were taken.

Figure 1 Lafayette manual muscle tester.

Subjects’ jump measurements occurred in a separate room by two other examiners to maintain examiner blindness from the strength and girth measurements. These examiners were also blind to group assignment. For the vertical jump, subjects stood in one spot, feet shoulder width apart, and jumped as high as they could while coming off the ground with both feet simultaneously, and landing on both feet together. A smartphone application called “My Jump 2” was used to measure the vertical jump. This application has demonstrated high reliability (ICC = 0.96) and high correlation to force plate jump height measurement (r = 0.96) in a previous study (Haynes et al., 2018). Subjects were provided a practice jump, and then three trials were performed for data collection. For the broad jump (horizontal jump), subjects stood behind a line with their toes at the edge and feet shoulder width apart, jumped as far as they could forward, and landed the jump without extra footsteps. Measurements were taken from the heel closest to the line from which the subjects began the jump. This task has also demonstrated measurement reliability (ICC = 0.86) in a previous study (Pehar et al., 2017). Subjects were provided a practice jump, and then three trials were performed for data collection.

The last measurement was the timed single-leg bridge, taken at least 90 s following jump measurements to allow adequate rest. This measure has demonstrated high reliability (ICC = 0.87–0.99) with a low standard error of measurement (SEM = 8.9 s) (Lehecka, 2018). Subjects used their dominant leg (the leg with which they would kick a ball) for the single-leg bridge. Subjects were positioned in hooklying with the knee to be measured flexed to 135° as opposed to 90° to decrease hamstring activation and the chance of cramping (Lehecka et al., 2017). They crossed their arms across their chest and bridged up with both legs until the knees and hip were in line with the torso and shoulders, then they fully extended the knee of the non-dominant leg, keeping both thighs parallel, and held the position as long as possible. During the bridge, an inclinometer was used to ensure subjects remained within a five-degree variance of the starting position throughout the bridge in both the sagittal and frontal planes. The measurement was recorded in seconds.

Subjects were instructed to perform either seated gluteal squeezes (Fig. 2) or supine bilateral bridges (Fig. 3) based on their randomized group assignment for 15 min rather than a set number of repetitions each day during the 8-week intervention period. Repetitions of each exercise were instructed to take 5 s followed by a brief relaxation. Subjects could perform the exercises in either one or multiple daily bouts. Subjects were provided brief email reminders at the second, fourth, and sixth week marks encouraging compliance and compliance tracking. Post-intervention measures were taken identically to pre-intervention measures.

Figure 2 The gluteal squeeze exercise performed with the medial edges of the feet together and knees shoulder-width apart.

Figure 3 The bilateral bridge exercise in the raised position after rising from the surface.

Gluteal squeezes were prescribed to be performed in sitting versus standing for increased convenience. This position was also expected to lessen hamstring activation during the exercise compared to standing. Subjects in the squeeze group sat with the medial edges of their feet in contact with each other, their knees flexed approximately 90° and positioned shoulder-width apart, and their trunk upright. They were instructed to maintain this posture while performing a maximal bilateral isometric contraction of their gluteals. Subjects were advised to palpate the gluteals to ensure muscle activation. They were also to expect the isometric contractions to cause a slight rise in their seated position. Subjects could also monitor their hamstrings and quadriceps with palpation to ensure relative relaxation. The gluteal squeezes, held for 5 s each followed by a brief relaxation, were prescribed for a total of 15 min each day during the 8-week intervention period.

The supine bilateral bridges were prescribed for an identical daily time of 15 min as the gluteal squeezes, and the bridges were also instructed to take 5 s each (up and down from the ground representing one repetition). They included 135° of knee flexion bilaterally as the subjects lied supine in an attempt to limit hamstring activity (Lehecka et al., 2017). Subjects in the bridge group crossed their arms across their chest and raised their pelvis from the ground until the hips reached neutral during the exercise. Following a brief hold, subjects were instructed to lower down with a speed equal to that of the lifting phase. Footwear was not controlled.

Data analysis performed using SPSS v23.0 (SPSS Inc., Chicaco, IL, USA) included dependent and independent t-tests to determine significant changes between pre- and post-intervention measures or group differences in change scores. Descriptive statistics were calculated for all measures and demographic variables in addition to subjects’ exercise compliance. The researcher performing data analysis was blind to the intervention assignment of groups.

Results

A total of 30 subjects’ data were analyzed (mean age = 23.28 ± 2.15 years; mean height = 1.74 ± 0.10 m; mean weight = 70.97 ± 12.74 kg). One subject’s data were excluded after it was determined to be recorded incorrectly at baseline, and one subject was excluded due to a lower extremity injury separate from the study during the intervention period. Both excluded subjects were in the bridge group which, in turn, contained 14 subjects while the squeeze group contained 16 subjects.

Between and within group differences were examined. No statistically significant difference existed between the bridge and squeeze groups at baseline, and no statistical difference was found between groups for any measure after the 8 weeks of intervention. Compliance was 51.35% ± 19.38% for the bridge group, and 64.15% ± 26.16% for the squeeze group, measured as the percentage of days during which 15 min of the prescribed exercise was performed.

Both groups significantly improved hip extension strength. In the bridge group, the average increase from baseline in right and left gluteus maximus strength was 4.05 ± 5.10 kg (p = 0.011) and 3.84 ± 3.86 kg (p = 0.003), respectively. These values correspond to an average increase of 11.30% in gluteus maximus strength bilaterally in the bridge group. Subjects in the squeeze group increased right and left gluteus maximus strength by 5.75 ± 4.99 kg (p = 0.000) and 5.14 ± 4.35 kg (p = 0.000), respectively, which equate to an average 15.56% strength increase bilaterally over the 8 weeks of training. A complete list of the changes between baseline and post-intervention measures are shown in Table 1. Mean values for all baseline and post-intervention measures are shown in Table 2.

Table 1 Changes in strength, endurance, power, and girth outcomes over 8 weeks of intervention.

	Mean change ± SD for bridge group (95% CI) (n = 14)	p-value for bridge group changes	Mean change ± SD for squeeze group (95% CI) (n = 16)	p-value for squeeze group changes	
Right hip extension strength	4.05 ± 5.10 kg*
[1.11–7.00]	0.011*	5.75 ± 4.99 kg*
[3.10–8.41]	0.000*	
Left hip extension strength	3.84 ± 3.86 kg*
[1.61–6.07]	0.003*	5.14 ± 4.35 kg*
[2.82–7.46]	0.000*	
Single-leg bridge endurance	12.14 ± 36.92 s
[−9.18–33.46]	0.240	10.71 ± 30.00 s
[−5.27–26.69]	0.174	
Vertical jump	0.40 ± 2.47 cm
[−1.03–1.83]	0.555	0.22 ± 2.72 cm
[−1.67–1.23]	0.747	
Broad jump	1.51 ± 8.13 cm
[−4.85–4.55]	0.946	1.60 ± 13.18 cm
[−8.62–5.42]	0.635	
Girth five cm above greater trochanter	0.31 ± 4.0 cm
[−1.97–2.60]	0.771	−0.71 ± 1.44 cm
[−1.47–0.06]	0.069	
Girth at the level of greater trochanter	0.19 ± 2.06 cm
[−1.00–1.38]	0.742	1.24 ± 1.60 cm*
[0.39–2.09]	0.007*	
Girth five cm below greater trochanter	0.14 ± 1.57 cm
[−0.76–1.05]	0.739	−0.74 ± 1.56 cm
[−1.57–0.09]	0.076	
Notes:

CI, confidence interval; cm, centimeters; kg, kilograms; s, seconds; SD, standard deviation.

* Statistically significant change at p ≤ 0.05 from pre- to post-measures.

Table 2 Baseline and post-intervention means for strength, endurance, power, and girth outcomes.

	Baseline mean ± SD for bridge group (n = 14)	Post-test mean ± SD for bridge group (n = 14)	Baseline mean ± SD for squeeze group (n = 16)	Post-test mean ± SD for squeeze group (n = 16)	
Right hip extension strength	36.73 ± 7.76 kg*	40.78 ± 7.17 kg*	34.41 ± 6.34 kg*	40.16 ± 7.03 kg*	
Left hip extension strength	33.14 ± 6.81 kg*	36.98 ± 5.22 kg*	35.62 ± 6.09 kg*	40.77 ± 5.66 kg*	
Single-leg bridge endurance	82.07 ± 36.21 s	94.21 ± 41.98 s	91.28 ± 43.12 s	101.99 ± 43.78 s	
Vertical jump	39.62 ± 10.85 cm	39.22 ± 11.12 cm	39.28 ± 13.09 cm	39.50 ± 11.91 cm	
Broad jump	194.36 ± 47.35 cm	194.51 ± 45.51 cm	193.90 ± 51.44 cm	195.50 ± 47.91 cm	
Girth five cm above greater trochanter	95.18 ± 5.32 cm	95.49 ± 5.97 cm	95.53 ± 8.01 cm	94.83 ± 8.22 cm	
Girth at the level of greater trochanter	97.83 ± 5.81 cm	98.01 ± 6.09 cm	97.53 ± 8.01 cm*	98.78 ± 7.99 cm*	
Girth five cm below greater trochanter	99.76 ± 6.79 cm	99.90 ± 6.37 cm	100.75 ± 8.65 cm	100.01 ± 8.41 cm	
Notes:

cm, centimeters; kg, kilograms; s, seconds; SD, standard deviation.

* Statistically significant change at p ≤ 0.05 from baseline to post-test measure.

The squeeze group significantly increased gluteal girth at the level of the greater trochanter by 1.24 ± 1.60 cm (p = 0.007). Girth decreases were observed five cm above (0.71 ± 1.44 cm) and below (0.74 ± 1.56 cm) the greater trochanter in the squeeze group, but these decreases were not statistically significant (p = 0.069–0.076). The bridge group demonstrated no significant changes in gluteal girth (p = 0.739–0.771). Gluteal girth changes in both groups are shown in Fig. 4.

Figure 4 Average changes in gluteal girth.

*Statistically significant change from baseline (p ≤ 0.05); cm, centimeters; GT, greater trochanter.

The bridge and squeeze groups both increased single-leg bridge endurance (12.14 ± 36.92 s and 10.71 ± 30.00 s, respectively), but neither of these increases reached statistical significance (p = 0.174–0.240). No statistically significant change occurred for either jump task in either group (p = 0.555–0.946).

Discussion

The primary purpose of this study was to compare the effects of a common exercise for hip extension training—the supine bilateral bridge—to a more convenient and relatively unstudied exercise for the gluteus maximus, the seated gluteal squeeze. The main finding was that both groups significantly improved hip extension strength over 8 weeks of training (p = 0.000–0.011), and the squeeze group demonstrated more strength improvement than the bridge group (15.56% strength increase from baseline measures in the squeeze group compared to 11.30% in the bridge group). Hip extension strength measurements were taken in prone with the hip in 0° of hip extension, unlike the strengthening position of the gluteal squeezes in 90° of hip flexion. This provides some evidence that strengthening in one position can affect strength in alternate positions of the hip.

To the authors’ knowledge, no studies have examined the effects of these exercises over a specific training period, so comparison of this study’s results to others’ is limited, and this study fills a gap in current knowledge. Studies implementing a training program for hip muscles typically include multiple exercises during an intervention period, making it difficult to determine the effects of a single exercise (Bennell et al., 2010; French, Gilsenan & Cusack, 2008). For example, one 12-week study of progressive hip strengthening and other exercises performed during three 60-min sessions per week by subjects with hip osteoarthritis demonstrated a 20% increase in subjects’ isometric leg extension strength, but the independent effects of the squat, sit-to-stand, step board, knee flexion, balance, and other exercises employed are unknown (Uusi-Rasi et al., 2017). Nevertheless, strengthening targeting the gluteus maximus appears beneficial for the rehabilitation of several pathologies, including hip, knee, and low back pain (Ferber et al., 2015; Jeong et al., 2015; Uusi-Rasi et al., 2017). So, the supine bilateral bridge and seated gluteal squeeze used in this study may aid in the rehabilitation of these conditions. There is contention that hip extension strength measured via dynamometry may not directly translate or correlate well to specific functional tasks such as jumping as seen in this study. Nevertheless, hip extension strength does appear to be correlated with rehabilitation of the aforementioned pathologies.

No significant differences were seen by either group in the vertical and broad jump outcomes. The effects of other hip exercises on both vertical and broad jump distance have been examined, but these studies also typically include other training components that limit comparison to the current study. For example, one study of barbell hip thrust versus front squat training programs over 6 weeks found notable improvements in vertical jump (3.42–7.30%) and broad jump (1.71–2.38%) measures (Contreras et al., 2017). Those values are larger than the non-significant increases found in the current study. However, in addition to the barbell hip thrust and front squat exercises which were performed with progressive resistance starting at 60% of a three-repetition maximum, subjects were also prescribed upper-body and core exercises (Contreras et al., 2017). The lack of significant differences in jump measures in the current study may have been due to a lack of progressive resistance or a lack of training specificity. Plyometric programs and specific jump training have demonstrated statistically significant increases in these jump tasks (Marián et al., 2016; Markovic, 2007).

Single-leg bridge endurance increased in both the bridge group (12.14 s or 15.25%) and squeeze group (10.71 s or 11.46%). Although this outcome measure has not been used previously in an intervention study, similar average values were found among healthy subjects in the seminal study of the single-leg bridge endurance test (81.03 ± 24.79 s) as those for subjects in the current study (86.98 ± 38.97 s) (Lehecka, 2018).

Also notable in the current study was the statistically significant increase in gluteal girth at the level of the greater trochanter seen in the squeeze group, but not in the bridge group. Moreover, decreases in gluteal girth were seen five cm above (0.71 cm) and below (0.74 cm) the greater trochanter in the squeeze group, suggesting a rounding of the gluteal curve (Fig. 4). This rounding (girth increase at the level of the trochanter, and decreases above and below the trochanter) may be a result of a redistribution of muscle mass, perhaps due to changes in resting muscle tone, although this cannot be confirmed without imaging procedures. Fat loss above and below the trochanter is possible, albeit unlikely with the energy demands of this intervention. The p-values for the changes above and below the greater trochanter approached statistical significance (p = 0.069 and p = 0.076, respectively), but were not statistically significant at the p ≤ 0.05 level. Hip muscle thickness changes following training programs including the supine bilateral bridge or gluteal squeeze have not been performed to the authors’ knowledge prior to the current study. However, several studies have examined thigh muscle hypertrophy following resistance training (Schoenfeld et al., 2016a). For example, one study used ultrasound imaging of the lateral thigh before and after an 8-week training study including performance of the barbell back squat, machine leg press, and machine leg extension three times per week (Schoenfeld et al., 2016b). Results from that study found changes in lateral thigh muscle thickness between 2.3 and 5.8 mm depending on whether the group performed moderate or heavy resistance exercises. Ultrasound imaging is arguably a more valid measure of specific muscle thickness changes, despite being more time-consuming and demanding more expensive equipment than the use of a tape measure. Nevertheless, girth measurements around the hip demonstrated similarly high reliability in the current study (ICC = 0.99).

The secondary purpose of this study was to compare the exercise compliance between groups. The 64.15% and 51.35% compliance results in the squeeze and bridge groups, respectively, were not statistically different, but may still be clinically meaningful. The gluteal squeeze does not require a subject to transfer to the floor or a bed to perform as does the supine bilateral bridge. This increased convenience of the seated gluteal squeeze may contribute to the higher degree of hip extension strength gain seen in the squeeze group compared to the bridge group. The compliance percentages equate to an average of about 9.6 min of training per day in the squeeze group, and 7.7 min of training per day in the bridge group given the prescription of 15 min of daily training.

Although the current study recruited healthy young adults, these compliance data may be of clinical importance for patients with hip pathology such as hip osteoarthritis and hip fracture. In one study of patients with hip and/or knee osteoarthritis, 53.8% of patients were adherent to the recommended activities at a 3-month follow-up, and adherence was positively associated with the effectiveness of the exercise therapy including pain outcomes and function (Pisters et al., 2010). A study of female patients after hip fracture concluded that strengthening self-efficacy and decreasing the fear of falling may improve adherence to exercise sessions (Resnick et al., 2008). Several studies support the idea that fear of falling is significantly associated with reduced levels of participation in physical activity in older adults (Bruce, Devine & Prince, 2002; Fletcher & Hirdes, 2004). This is notable in light of the current study because exercise adherence was higher among subjects in the squeeze group compared to the bridge group, and the seated gluteal squeeze would not require frequent transfers from the sitting position to perform, potentially minimizing the fear of falling barrier to exercise in an older population.

Several limitations of the current study exist. First, only healthy young adults were recruited, limiting the application of findings to other populations. Second, gluteal girth was assessed using a measuring tape around the circumference of the pelvis/hips instead of with the gold standard of ultrasound imaging directly over the gluteal masses. The girth measurements demonstrated high intra-rater reliability, and the measuring tape is more cost-effective than ultrasound imaging, but the validity of such measures was likely lacking in comparison to the gold standard. Third, subjects in the bridge group were not instructed to maximally squeeze their gluteals at the top of the movement, a cue used by some to potentially increase gluteal activation. Fourth, the training of subjects beyond the exercises prescribed for intervention was not controlled; however, this limitation affected both groups. If exercise and other forms of physical activity beyond the study interventions were restricted, confidence in the study interventions causing the study results instead of outside activities could be higher.

Conclusion

The findings of this study show that gluteal squeezes are as effective as bilateral bridges for significantly increasing hip extension strength, and both exercises can significantly increase hip extension strength over an 8-week training period. Gluteal squeezes also significantly increased girth at the level of the greater trochanter, although the supine bilateral bridges had no significant effect on girth. Subjects in the squeeze group demonstrated a higher exercise compliance than subjects in the bridge group, but this difference was not statistically significant. These results provide clinical and aesthetic reasons to perform gluteal squeezes. Future study should examine the effects of these interventions on subjects with hip, knee, or low back pathology.

Supplemental Information

Supplemental Information 1 Raw Data.

Click here for additional data file.

Additional Information and Declarations

Competing Interests

Author Contributions

Human Ethics

Data Availability

The authors declare that they have no competing interests.

Bryan J. Lehecka conceived and designed the experiments, performed the experiments, analyzed the data, contributed reagents/materials/analysis tools, prepared figures and/or tables, authored or reviewed drafts of the paper, approved the final draft.

Jessica Turley conceived and designed the experiments, performed the experiments, authored or reviewed drafts of the paper.

Aaron Stapleton conceived and designed the experiments, performed the experiments, authored or reviewed drafts of the paper.

Kyle Waits conceived and designed the experiments, performed the experiments, authored or reviewed drafts of the paper.

John Zirkle conceived and designed the experiments, performed the experiments, authored or reviewed drafts of the paper.

The following information was supplied relating to ethical approvals (i.e., approving body and any reference numbers):

The Wichita State University Institutional Review Board granted institutional approval to carry out the study within its facilities (WSU IRB #3875).

The following information was supplied regarding data availability:

The raw data is available as a Supplemental File.

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
