# Peer review of "The effects of gluteal squeezes compared to bilateral bridges on gluteal strength, power, endurance, and girth"

_PeerJ, doi:10.7717/peerj.7287_

## Round 0.1 · original submission · Minor Revisions

I apologize for the time it took to have reviews of your manuscript. The reviewers are very positive, and I agree with them. Please take their suggestions in full consideration.

·

Basic reporting

Make the citations like this (Contreras et al., 2015). The parentheses go before the period.

Every time you say, "in previous study," instead say "in a previous study."

Experimental design

I see why you chose the seated position for the squeeze group (to increase compliance), but this position of hip extension leads to lower glute activation compared to full hip extension (see Worrell et al., 2001). In addition, a narrow stance likely leads to reduced glute activation compared to a wider stance. This could mean that better results could have been achieved with standing glute squeezes if compliance was similar. Might want to mention this.

In addition, you didn't instruct the bridge group to squeeze their glutes maximally at the lockout portion of the movement. This may have evened out the results had you done so. May want to mention this as well.

Validity of the findings

Interestingly, squeezes which were in hip flexion transferred better to dynamometer strength measured in hip hyperextension compared to bridges which reach full hip extension. Might want to mention this.

Why would glute girth decrease at the measurements above and below the greater trochanter? It couldn't be muscle atrophy. But fat loss seems unlikely. Please offer some explanation.

Additional comments

The squeezes you mention in the intro were from a standing position which would involve a higher glute activation compared to a seated squeeze. Might want to mention this.

Page 6, line 53, change to: Gluteal girth is a less researched feature of the muscles compared to their strength, power, and endurance, but noteworthy nonetheless.

Page 7, line 92, change to: Hip girth measured similarly with a tape measure demonstrated high reliability (ICC = 0.96-0.97) in a previous study.

Page 8, line 117: I was under the impression that if someone was right handed, their left hip hip extensor is stronger while their right hip flexor was usually stronger (since they kick with their dominant leg). So you might have used the nondominant leg for testing. Could be wrong though. Just thought you might want to look into this.

Page 8, line 119: you mean as opposed to 90 degrees of knee flexion? If so, please state this.

Page 10, line 201, change to: For example, one 12-week study of progressive hip strengthening and other exercises

Page 11, line 255, change to: be of clinical importance

Page 12, line 275, please revise...I can't make sense of it

Reviewer 2 ·

Basic reporting

Please see entire review in comments

Experimental design

Please see entire review in comments

Validity of the findings

Please see entire review in comments

Additional comments

Congratulations to the authors on the production of this work. While this is a relatively simple study there is value in the answering of simple research questions. I have a few minor suggestions and questions. Below, I have quoted a part of your manuscript and made my comment.


Abstract

“Hip extension weakness is correlated with low back”

This is minor but it should be noted that hip extension weakness is only sometimes correlated with pathology/painful states. There are a number of papers where increased hip extension strength is correlated with pathology. This can be easily seen in the SR by Neal et al (2019) where hip Abduction strength was not related to PFPS and where increased hip strength was related to increased PFPS risk (https://www.ncbi.nlm.nih.gov/pubmed/30242107)


Introduction

Line 38-41 “For example, gluteal strength (hip abduction, external rotation, and extension strength) has independently predicted non-contact anterior cruciate ligament injuries and been associated with knee overuse injuries. (Khayambashi, 2016; Kollock, 2016)”

I realize the authors used the term “predicted” but this is again a bit strong. They calculated ORs after the fact. They didn’t identify people before hand and “predict” who got injured. Again, the point here is that we should be cautious in how certain we are when making the case for the relevance of hip strength to pathology. Yes, there are cases where there is an association but there are many papers where it is not associated.


Lines 93-95

We need a bit more information here. What was the make of the dynamometer? Was it peak force that was found? Was it the average force over 1 second etc? Can you add a picture of the set up. The methods should allow for the reproduction of this paper.

Line 104

Was the experimenter blinded to what stage the participant was in the study (e.g. pre or post) and blinded to group when using My Jump. My Jump requires a subjective determination of leaving the ground and landing. If you use this in future work I would suggest this blinding and perhaps having multiple assessors.

Line 126

I am actually having trouble seeing how many gluteal squeezes were prescribed here. The contraction lasted 5 seconds, they the rested briefly and were supposed to do this for 15 minutes? But, the 15 minutes could be split throughout the day? If someone did 5 seconds on and 2 seconds off they might perform around 8 repetitions per minute and could theoretically perform 15 minutes of this? Sixty 5 second glute squeezes in 15 minutes?

Was any of this tracked?

Last, pictures of both exercises would be helpful.


Line 167

How was compliance measured? Is this a percentage of days performed, percentage of the potential maximum repetitions, percentage of participants who did at least one set?

Line 251

The largest limitation of this study is the lack of a control group who did nothing. Both peak strength and endurance can be influenced by many factors. This should be noted.

It might also be relevant to note how poorly hip extension strength when tested with dyno poorly translates to functional tasks. One interpretation of your findings is not just that a Glute Squeeze performs similarly to a Glute Bridge but how poor of an exercise a Glute Bridge might be. Especially considering the prescription dosage of the exercise. Daily performance of low force, high repetitions (60) would not be the ideal way to increase strength.

Perhaps, adding a discussion on this can put your findings into context for the reader.

·

Basic reporting

No comment

Experimental design

129 the text reads that each group could perform the 15 minutes/day however they wanted which describes the "duration" of the daily exercise.
144 Text states that the bridge group did the same "frequency" as the squeeze group but that was never defined and was self selected in the squeeze group.

138 The squeeze group was instructed to do a MIVC of the glutes
144-151 Were the bridge subjects instructed to also use a maximal glute squeeze while doing the bridges or only whatever it took to lift the pelvis off the ground?

Validity of the findings

169-177 was there a statistically significant difference between the 11.3% increase in the bridge group and the 15.56% in the squeeze group?

178-183 do the authors have any theories as to why the girth increased with the squeeze group and not in the bridge group despite strength increases in both groups? How would they explain the strength increases without the hypertrophy found in the squeeze group?


222-226 Were these differences between groups statistically significant?

245-253 was there a follow-up questionnaire of the study subjects asking them about specific barriers they faced that resulted in the lower compliance in the bridge group. Was it the change of position required to complete the exercise or other circumstances?

Additional comments

Nice study, very well thought out.

---

## Round 0.2 · accepted · Accept

Thank you for your consideration of all reviewers' suggestions.

Please note that in line 47 you stressed: "pain seems more unlear (Neal et al., 2019)..." did you mean unclear?

Also in line 109 the word bling should be changed to blind?

If these details are taking into account, we will be ready to move on.